# Design thinking teaching and learning in higher education: Experiences across four universities

**Jacqueline E. McLaughlin**[1]*, **Elizabeth Chen**[2,3], **Danielle Lake**[4], **Wen Guo**[5], **Emily Rose Skywark**[3], **Aria Chernik**[6], **Tsailu Liu**[7]

**1** Division of Practice Advancement and Clinical Education, Center for Innovative Pharmacy Education and Research, UNC Eshelman School of Pharmacy, University of North Carolina at Chapel Hill, Chapel Hill, NC, United States of America, **2** Department of Health Behavior, UNC Gillings School of Global Public Health, University of North Carolina at Chapel Hill, Chapel Hill, NC, United States of America, **3** Office of the Vice Chancellor for Innovation, Entrepreneurship, and Economic Development, University of North Carolina at Chapel Hill, Chapel Hill, NC, United States of America, **4** Center for Design Thinking, Elon University, Elon, NC, United States of America, **5** Department of Art, Elon University, Elon, NC, United States of America, **6** Social Science Research Institute and Innovation and Entrepreneurship Initiative, Duke University, Durham, NC, United States of America, **7** Department of Graphic Design and Industrial Design, College of Design, North Carolina State University, Raleigh, NC, United States of America

* Jacqui_mclaughlin@unc.edu

**Data Availability Statement:** All data files are available from the Open-ICPSR database (https://doi.org/10.3886/E151681V1).

## Abstract

A growing body of literature highlights the increasing demand on college graduates to possess the problem finding, problem framing, and problem-solving skills necessary to address complex real-world challenges. Design thinking (DT) is an iterative, human-centered approach to problem solving that synthesizes what is desirable, equitable, technologically feasible, and sustainable. As universities expand efforts to train students with DT mindsets and skills, we must assess faculty and student DT practices and outcomes to better understand DT course experiences. Understanding how DT is taught and experienced within higher education can help schools promote student learning and align their training programs with professional, personal, and civic needs. In this study, surveys were completed by 19 faculty and 196 students from 23 courses at four universities. DT teaching and learning was characterized by three DT practices and five outcomes. Statistically significant differences were found by discipline of study and student type (i.e., graduate vs undergraduate), but not by gender or race/ethnicity. These results can be used to inform the development of classroom-based DT teaching and learning strategies across higher education institutions and disciplines.

## Introduction

Universities have faced considerable scrutiny in recent years for their apparent failure to adequately equip students with the complex reasoning and problem solving skills thought to be at the core of higher education [1–3]. A growing body of literature highlights the demand on

**Funding:** The author(s) received no specific funding for this work.

**Competing interests:** The authors have declared that no competing interests exist.

college graduates to simultaneously master the disciplinary knowledge and mindsets necessary to address complex real-world problems [4–7]. These demands, coupled with ongoing concerns about the quality of higher education, have drawn attention to the need to rethink our focus within higher education [8–11].

Design thinking (DT) is an iterative, creative approach to problem finding, problem framing, and problem solving that synthesizes what is desirable to real stakeholders, equitable, technologically feasible and sustainable [12, 13]. Most models of design thinking move through (1) inspiration, empathy and problem definition, (2) ideation, (3) prototyping and testing and (4) implementation stages [14]. By beginning with the goals and needs of stakeholders and engaging in short iteration cycles, DT supports collaborative solutions that roll out with lasting impact [12, 15]. Professional design consultancies often use this method to design innovative products or services. Thus, DT is a tool frequently taught in business, engineering and design schools.

A growing number of disciplines are utilizing and teaching DT to solve complex problems, including public health, healthcare, and the liberal arts [12, 16–18]. The success of innovations developed with the DT process has led to the uptake of DT to solve various challenges, including customer experience and strategic planning, and to support various sectors, including government agencies, non-profits, educational institutions, and community organizations [15]. Power dynamics can be challenged at the intersection of design for social innovation; this form of participatory design can help students imagine new ways of thinking to solve complex problems [19]. The human-centered, real-world solutions generated from the DT process have the potential to provide more systemic solutions to difficult social problems, like climate change, poverty, housing instability, and health promotion.

As DT is adopted by a broader audience, there is an onus on educators to equip students across university disciplines with tools and mindsets valuable for addressing these complex, real-world problems. This includes, but is not limited to "situatedness", self-reflection, empathetic listening, critical observation and creative collaboration [20–23]. DT pedagogy should (1) frame both the situation and the student's place within the situation; 2) allow for iterative exploration across space and time alongside diverse stakeholders 3) require the generation of divergent possibilities 4) the prototyping and actionable testing of these possibilities and (5) develop sustainable commitments to cultivated change [21].

Within higher education, we see new centers, programs, and courses being established outside common DT fields (i.e., business, engineering, design) with a focus on teaching DT mindsets and skills, such as Tulane University's Phyllis M. Taylor Center for Social Innovation, University of Illinois' Seibel Center for Design, and Design Thinking and Elon University's Center for Design Thinking. As universities expand efforts to train students with DT mindsets and skills, we must assess faculty and student DT practices and outcomes to better understand DT course experiences. In one single institution study, for example, researchers found that DT requires time and trust which can be constrained by the imposed deadlines of semester-based projects [21]. In a single course study, students indicated that their "whirlwind" course promoted almost "exponential" growth [22]. While survey instruments that measure DT practices and outcomes have been validated across diverse workplace settings [24], the assessment of DT practices and outcomes across higher education is still new.

DT pedagogy in higher education is incompletely understood, particularly in fields beyond traditional design disciplines (i.e., de-disciplined design). We expanded a single institution study of faculty by Lake and Colleagues [21] in an effort to assess DT teaching and learning (DT-TL) experiences of undergraduate students, graduate students, and faculty at four universities within a southeastern state of the United States. Overall, we set out to answer the

following research question: how do faculty and students experience design thinking within higher education courses? To answer this question, we explored the following subquestions:

1. What kinds of DT practices are experienced within higher education courses? In what ways do higher education DT practices align with and differ from other industries?

2. What kinds of outcomes are experienced within higher education courses? In what ways do outcomes of DT in higher education align with and differ from other industries?

3. Is DT a valid construct within higher education teaching and learning?

4. What differences exist between groups, such as discipline (business, design & engineering, other), and student type (undergraduate, graduate)?

## Methods

To answer these research questions, we utilized a mixed methods design that combined faculty and student surveys with semi-structured interviews. This manuscript focuses on the survey data only. Additional qualitative and mixed methods findings are detailed elsewhere [25].

### Survey development

The survey used in this study was adapted from Liedtka and Bahr, who studied DT practices and outcomes among employees of for-profit, non-profit, and government sectors [24]. Liedtka and Bahr survey items were adapted to align with the context of higher education and additional items were included based on research team experiences. The final survey included 11 items about DT practices, 42 items about outcomes from DT, 11 demographic items (e.g., *What is your gender identity*?), and 11 course descriptor items (e.g., *Was the term "design thinking" explicitly referenced in the course*?). All DT practices and outcomes items were prompted by the following stem: *Please note how often, as a direct result of this specific course, you observed the following [practices/outcomes]* and measured on a scale from 1-Never to 5-Almost Always. The research team reviewed the survey for face validity prior to data collection.

### Data collection

Students and faculty at four universities in the southeastern United States were recruited for the study during the 2020–2021 academic year. Purposive sampling was used to identify and recruit participants based on their experience with the research focus [26]. In August 2020, research team members recruited faculty at their home institutions who they knew taught design thinking courses. The email recruitment invited faculty to fill out an initial interest survey about their course, including: course number and title; number of students enrolled in their course; whether their course was open to undergraduate students, graduate students, or both; and number of credit hours associated with the course. The survey also asked faculty for informed consent, contact information, and whether they were interested in participating in an optional semi-structured interview at the end of the semester. Faculty who agreed to participate were also expected to recruit students from their DT course to the study.

Toward the end of the Fall 2020 semester, research team members emailed the faculty survey to consented faculty from their home institutions. Faculty participants were also asked to forward study invitations to students enrolled in their DT courses. The email invitations drafted by our research team included information about the study and a link to the student survey. Students interested in participating in the study provided informed consent in the online survey and were also asked whether they wanted to schedule an optional follow-on

interview. Faculty participants received three email reminders regarding study invitations to their students before the end of the Fall 2020 semester.

## Data analysis

Survey data were first analyzed using descriptive statistics, with continuous variables reported as mean ± standard deviation (SD) and categorical variables reported as frequency (percent). An exploratory factor analysis was conducted to identify DT teaching and learning (DT-TL) constructs using principal components analysis with varimax rotation and Kaiser rule (ie, eigenvalues < 1.0) for student survey items also included in the Liedka and Bahr study [24]. Bivariate correlations were calculated using Pearson rho ($r_p$) and reliabilities were calculated using Cronbach α. Group comparisons were examined using independent t-tests and one-way ANOVA with Bonferroni post hoc analysis. Parametric statistics were considered appropriate due to normality of data and sufficient sample size. Statistical significance was established at α<0.05. All data analysis was performed in SPSS for Windows, v26 (IBM, Armonk, NY).

## Ethical considerations, consent

This project was submitted to the UNC Institutional Review Board (#20–2316), Elon University Institutional Review Board (#21–031), Duke Campus Institutional Review Board (#2021–0168), and North Carolina State University (#23502). The submission was approved or determined to be exempt from further review by each review board according to 45 CFR 46.104. Written consent was obtained electronically from all participants at the start of the survey.

## Results

Surveys were completed by 19 faculty and 196 students from 23 courses at four universities. The response rate for faculty was 84.2%. Based on the number of students enrolled in each course, our estimated response rate of students was 20.8%. As seen in Table 1, student participants were predominately white (n = 132, 63.4%), female (n = 126, 64.2%), and majoring in Interdisciplinary Humanities/Social Sciences (n = 105, 53.6%). Eighty-seven percent (n = 170) of students were undergraduate students. Similarly, faculty participants were predominately white (n = 14, 73.7%), female (n = 11, 57.9%), and from Interdisciplinary Humanities/Social Sciences (n = 11, 57.9%).

When asked about the course, most students (n = 149, 76.0%) and faculty (n = 14, 73.7%) indicated that DT was explicitly taught in the course and that they utilized a real-world project (n = 160 (81.6%) students and 13 (68.4%) faculty). On average, students reported spending 69.69% ± 21.65% of their time working in teams. Most students indicated having none or limited DT expertise *prior to the course* (n = 146, 74.4%) and moderate or extensive expertise *after the course* (n = 152, 77.5%). Similarly, the number of faculty reporting none (n = 0, 0%) or limited DT expertise (n = 7, 36.8%) *prior to the course* dropped to 2 (10.5%) *after the course*.

Table 2 provides item-level responses for DT practices in the current study and two related DT studies using the same survey items. On a five-point scale from 1-Never to 5-Almost Always, student and faculty participants reported that they used all 11 of the DT practices with moderate to high frequency in the course. Faculty indicated that they most commonly *followed a structured process* (4.16 ± 0.69), *created prototypes of ideas* (4.11 ± 0.66), and *emphasized active listening among team to find shared meaning* (4.11 ± 0.88). Similarly, students most commonly *emphasized active listening among team to find shared meanting* (4.38 ± 0.77), *followed a structured process* (4.16 ± 0.75), and *generated a diverse set of ideas* (4.16 ± 0.85). Faculty and students *executed real world experiments* least frequently (3.16 ± 1.26 and 2.96 ± 1.34, respectively).

**Table 1. Characteristics of survey participants and courses.**

| Participant Characteristics | Students (n = 196) | Faculty (n = 19) |
| --- | --- | --- |
| **Race:** *White* | 132 (63.4%) | 14 (73.7%) |
| *Black/African American* | 14 (7.1%) | 0 (0%) |
| *Asian* | 21 (10.7%) | 4 (28.6%) |
| *Hawaiian/Pacific Islander* | 5 (2.6%) | 0 (0%) |
| *American Indian/Alaska* | 2 (1.0%) | 0 (0%) |
| **Ethnicity:** *Hispanic* | 10 (5.6%) | 0 (0%) |
| **Gender:** *Female* | 126 (64.2%) | 11 (57.9%) |
| **Discipline:** *Business* | 32 (16.3%) | 4 (21.1%) |
| *Design & Engineering* | 45 (23.0%) | 4 (21.1%) |
| *Interdisciplinary Humanities/Social Sciences* | 105 (53.6%) | 11 (57.9%) |
| **Undergraduate Student:** *Yes* | 170 (86.7%) | N/A |
| **DT Expertise Prior to Course:** *None* | 51 (26.0%) | 0 (0%) |
| *Limited* | 95 (48.4%) | 7 (36.8%) |
| *Moderate* | 26 (13.3%) | 7 (36.8%) |
| *Extensive* | 5 (2.6%) | 4 (21.1%) |
| **DT Expertise After the Course:** *None* | 0 (0%) | 0 (0%) |
| *Limited* | 25 (12.8%) | 2 (10.5%) |
| *Moderate* | 118 (60.2%) | 12 (63.2%) |
| *Extensive* | 34 (17.3%) | 4 (21.1%) |
| **Course Characteristics** | | |
| **DT Explicitly Mentioned:** *Yes* | 149 (76.0%) | 14 (73.7%) |
| **Had Resources Needed for DT:** Agree | 143 (72.9%) | 9 (47.7%) |
| **Space Was Conducive to DT:** Agree | 100 (51.0%) | 9 (47.7%) |
| **Real World Project used for DT:** *Yes* | 160 (81.6%) | 13 (68.4%) |
| **Percent of Course Time Spent in Teams** (mean ± SD) | 69.69±21.65% | 66.89±26.14% |

DT = Design Thinking; SD = Standard Deviation; N/A = Not Applicable

*NOTE*: some variables have missing data; percentages may sum to less than 100%

As seen in Table 3, factor analyses indicated that DT-TL practices can be broadly characterized by three constructs accounting for 59.03% of the variance: *Discovery and Ideation* (4.04 ± 0.70, 21.67% of variance); *Team Formation and Functioning* (4.21 ±0.53, 16.02% of variance); and *Prototyping and Experimentation* (3.44 ±0.94, 21.34% of variance). Survey items loaded into the same three factors as Liedtka and Bahr [24], with the exception of "Followed a Structured Process," which loaded to *Discovery and Ideation* in Liedkta and Bahr (2019) instead of on *Team Formation and Functioning* in our study. Students who reported that DT was explicitly taught in their course more frequently engaged in *Discovery and Ideation* than those who indicated DT was not explicitly taught (4.16±0.65 vs 3.42±0.72, p<0.001). Students from Business and Design & Engineering disciplines also more frequently engaged in *Discovery and Ideation* than those from Interdisciplinary Humanities/Social Sciences (4.21±0.53 vs 3.85±0.69, p = .04; 4.26±0.74 vs 3.85±0.69, p = 0.004 respectively). Undergraduate students reported *Prototyping and Experimentation* more frequently than graduate students (3.49±0.88 vs 3.07±1.24, p = 0.04). There were no differences in DT practices found by race/ethnicity, or gender.

Table 4 provides factors loadings for outcomes of DT in higher education courses, which are broadly characterized by five constructs accounting for 63.00% of the variance: *Implementation Support* (4.00 ± 0.70, 18.51%), *Psychological Benefits and Motivation* (4.15 ± 0.64,

Table 2. Survey responses for DT practices in current study (faculty, students) and in Lake et al. [21] and Liedtka and Bahr [24].

| Design Thinking Practice<br><br>*Please note how often, as a direct result of this specific course, you observed the following practices:* | Faculty (n = 19) | Students (n = 196) | Lake, et al. [21]<br><br>(n = 35) | Liedtka & Bahr [24] (n = 416) |
|---|---|---|---|---|
| | Mean (SD) | Mean (SD) | Mean (SD) | Mean (SD) |
| 1. Followed a structured process | 4.16 (0.69) | 4.16 (0.75) | --- | 3.80 (0.86) |
| 2. Formed a diverse team | 3.74 (0.99) | 4.09 (0.89) | 4.09 s(---) | 3.93 (0.87) |
| 3. Emphasized active listening among team to find shared meaning | 4.11 (0.88) | 4.38 (0.77) | 3.80 (---) | 4.01 (0.95) |
| 4.Done user research using ethnographic tools | 3.63 (1.21) | 3.89 (1.13) | 2.69(---) | 3.91 (1.07) |
| 5. Focused your problem definition on user's perspective rather than the organization's | 4.32 (0.82) | 4.12 (0.78) | 3.00(---) | 4.09 (0.88) |
| 6. Created a set of design criteria that described an ideal solution based on user research | 3.34 (1.30) | 3.99 (0.89) | 3.11(---) | 3.60 (1.03) |
| 7. Generated a diverse set of ideas based on your user research | 3.79 (0.98) | 4.16 (0.85) | --- | 3.90 (0.95) |
| 8. Created prototypes of your ideas | 4.11 (0.66) | 3.86 (1.10) | 3.27(---) | 3.80 (1.05) |
| 9. Moved multiples ideas into prototyping and testing | 3.26 (0.93) | 3.22 (1.19) | 2.74(---) | 3.39 (1.02) |
| 10. Got feedback from users and other stakeholders on the prototype | 3.68 (1.25) | 3.73 (1.24) | 2.51(---) | 3.85 (1.05) |
| 11. Executed real world experiments to test your ideas | 3.16 (1.26) | 2.96 (1.34) | 2.71(---) | 3.43 (1.07) |
| *Cronbach Alpha* | 0.89 | 0.81 | --- | --- |

**SD** = Standard Deviation; All items measured on a scale from 1-Never to 5-Almost Always

"—" indicates item was not included on survey and/or value was not reported.

14.13%), *Relationships and Trust* (3.76 ±0.82, 12.74%), *Quality of Solutions Generated* (4.21 ± 1.75, 8.98%), and *Individual Adaptation and Flexibility* (4.13 ± 0.66, 8.64%). Students who reported that DT was explicitly taught more frequently experienced *Psychological Benefits and Motivation* than those who indicated DT was not explicitly taught in their course (4.21 ±0.62 vs 3.85±0.87, p = 0.008). There were no differences in outcomes of DT found by race/ethnicity, gender, or student type (ie, undergraduate vs graduate).

As seen in Table 5, four of the five outcomes factors contained similar items and items that differed. Factor 3, for example, emphasized *Relationships* in both studies and included the survey item *Built new relationships locally that continued after the initial project was completed*. However, in the current study, other items in that factor addressed *Trust* (e.g., *Built trust among team members)* while other items in the Liedtka and Bahr study addressed *Resources* (e.g., *Expanded access to new resources for individuals and teams*.)

All correlations between the three DT practice and five outcome constructs were statistically significant, ranging from positive, moderate relationships ($r_p$ = 0.33) to positive, very strong relationships ($r_p$ = 0.76) (Table 6). Cronbach alpha exceeded .6 for seven of the eight DT constructs ($0.53 \leq \alpha \leq 0.89$), suggesting that the items used to create each construct demonstrated acceptable internal consistency.

## Discussion

This study explored faculty and student experiences with DT in courses from various disciplines within four universities. Given the increasing uptake of DT in higher education and across professional fields, this study is timely and critical for understanding the types of DT practices and outcomes experienced, and ways in which DT in these settings might vary among different industries and stakeholders. This work extends research conducted in workplace settings [24] and compliments studies exploring conceptual frameworks and uses of DT in education [27, 28]. Luka, for example, integrated multiple DT models to design an

**Table 3. Factor loadings and group differences for DT practices experienced in higher education courses (n = 196).**

| DT Practice | Factor | | |
|---|---|---|---|
| | Discovery & Ideation | Team Formation & Functioning | Prototyping & Experimentation |
| Followed a structured process | | 0.46 | |
| Formed a diverse team | | 0.75 | |
| Emphasized active listening among team to find shared meaning | | 0.70 | |
| Done user research using ethnographic tools | 0.67 | | |
| Focused your problem definition on user's perspective rather than the organization's | 0.75 | | |
| Created a set of design criteria that described an ideal solution based on user research | 0.73 | | |
| Generated a diverse set of ideas based on your user research | 0.69 | | |
| Created prototypes of your ideas | | | 0.69 |
| Moved multiples ideas into prototyping and testing | | | 0.75 |
| Got feedback from users and other stakeholders on the prototype | | | 0.81 |
| Executed real world experiments to test your ideas | | | 0.70 |
| Factor Mean (SD) | 4.04 (0.70) | 4.21 (0.58) | 3.44 (0.94) |
| **Race/Ethnicity:** *Underrepresented** | 4.06 ± 0.74 | 4.06 ± 0.75 | 3.30 ± 0.87 |
| *Other* | 4.04 ± 0.71 | 4.25 ± 0.55 | 3.51 ± 0.94 |
| **Gender:** *Female* | 4.09 ± 0.70 | 4.22 ± 0.59 | 3.45 ± 0.96 |
| *Male* | 3.95 ± 0.73 | 4.24 ± 0.58 | 3.56 ± 0.80 |
| **Discipline:** *Business* | 4.21 ± 0.53** | 4.29 ± 0.60 | 3.65 ± 1.08 |
| *Design & Engineering* | 4.26 ± 0.74** | 4.19 ± 0.56 | 3.40 ± 0.87 |
| *Interdisciplinary Humanities/Social Sciences* | 3.85 ± 0.69** | 4.15 ± 0.59 | 3.43 ± 0.92 |
| **Student Type:** *Undergraduate Students* | 4.03 ± 0.70 | 4.22 ± 0.58 | 3.49 ± 0.88** |
| *Graduate Students* | 4.06 ± 0.70 | 4.19 ± 0.58 | 3.07 ± 1.24** |

SD = Standard Deviation; DT = Design Thinking

Principal Component Analysis with Varimax rotation using Kaiser Normalization converged in 5 iterations and accounted for 59.03% of variance.

*Underrepresented race/ethnicity includes Black/African American, Hawaiian/Pacific Islander, American Indian/Alaskan, and Hispanic

**$p < 0.05$ for *Discovery and Ideation* (Business and Design & Engineering disciplines more frequently engaged than Interdisciplinary Humanities/Social Sciences) and *Prototyping and Experimentation (*Undergraduate students more frequently engaged than graduate students).

international English course that engaged learners in a four-phased learning cycle of experiencing, reflecting, thinking and acting [27] while Wrigley and Straker described the Educational Design Ladder, which illustrates the organization of a multidisciplinary DT program [28].

## DT-TL constructs

The results of this study suggest that DT-TL is a valid construct in its own right within the context of higher education. Specifically, the factor analysis revealed 8 distinct factors with high factor loads and the majority of variance accounted for by the analysis, providing support for content specificity. While the 3 DT-TL practice constructs and 5 outcomes constructs are aligned with those described in other contexts [24], they also embody items differently, which may be attributable to the different processes and contexts associated with student learning in higher education environments. As noted below, the experiences of students and faculty in DT-TL–as indicated by item ratings and construct scores–are likely influenced by interactions with others, constraints of higher education systems (e.g., semester timelines), and disciplinary

**Table 4. Factor loadings and group differences for outcomes of DT in higher education courses (n = 196).**

| Outcome | Factor | | | | |
|---|---|---|---|---|---|
| | **Implementation Support** | **Psychological Benefits & Motivation** | **Relationships &Trust** | **Quality of Solutions Generated** | **Individual Adaptation & Flexibility** |
| Helped me see the problems | | | | | 0.76 |
| Enhanced my ability to pivot when initial solutions didn't work | | | | | 0.76 |
| Built new relationships locally that continued after the initial project was completed | | | 0.71 | | |
| Expanded access to new resources for individuals and teams | 0.63 | | | | |
| Helped pool resources for greater impact | | | | | |
| Enhanced other stakeholders willingness to collaborate on new solutions | | | 0.72 | | |
| Built trust among team members | | | 0.48 | | |
| Built trust between problem-solving teams and other stakeholders | | | 0.70 | | |
| Allowed new and better solutions, not visible at the beginning of the process, to emerge during it | | | | 0.62 | |
| Fostered the inclusion of user input | | | | 0.79 | |
| Helped people involved to examine their own biases and preconceptions | | | 0.52 | | |
| Created a sense of safety to try new things | 0.58 | | | | |
| Gave people more confidence in their own creative abilities | | 0.44 | | | |
| Improved the likelihood of the implementation of new solutions | 0.47 | | | | |
| Made it easier to discard solutions that didn't work as planned | | | | | 0.44 |
| Helped people interested in trying new things to connect and support each other | 0.70 | | | | |
| Encouraged people's open-mindedness to try new things | 0.60 | | | | |
| Encouraged shifts in organizational culture that made it more customer-focused | 0.66 | | | | |
| Encouraged changes in organizational culture that made risk-taking more acceptable | 0.75 | | | | |
| Kept people motivated to work on a project to achieve impact | | 0.58 | | | |
| Broadened organization's definition of what innovation is | 0.45 | | | | |
| Increased a sense of ownership and acceptance of a solution | | 0.72 | | | |
| Increased appreciation for use of data to help drive decisions | | 0.62 | | | |
| Increased engagement of teammates involved in the design thinking process | | 0.77 | | | |
| **Factor Mean (SD)** | **4.00 (0.70)** | **4.15 (0.64)** | **3.76 (0.82)** | **4.21 (0.75)** | **4.13 (0.66)** |
| **Race/Ethnicity:** *Underrepresented** | 3.89 ± 0.76 | 4.14 ± 0.51 | 3.59 ± 0.72 | 4.28 ± 0.66 | 4.05 ± 0.73 |
| *Other* | 4.03 ± 0.70 | 4.15 ± 0.66 | 3.79 ± 0.83 | 4.19 ± 0.78 | 4.17 ± 0.64 |
| **Gender:** *Female* | 4.03 ± 0.72 | 4.19 ± 0.63 | 3.77 ± 0.86 | 4.26 ± 0.76 | 4.14 ± 0.68 |
| *Male* | 3.96 ± 0.67 | 4.09 ± 0.64 | 3.73 ± 0.69 | 4.08 ± 0.72 | 4.17 ± 0.60 |
| **Discipline:** *Business* | 3.98 ± 0.87 | 4.14 ± 0.74 | 3.57 ± 0.99 | 4.23 ± 0.64 | 4.22 ± 0.65 |

*(Continued)*

**Table 4.** (*Continued*)

| Outcome | Factor | | | | |
|---|---|---|---|---|---|
| | **Implementation Support** | **Psychological Benefits & Motivation** | **Relationships &Trust** | **Quality of Solutions Generated** | **Individual Adaptation & Flexibility** |
| *Design & Engineering* | 3.89 ± 0.61 | 4.15 ± 0.54 | 3.52 ± 0.79 | 4.21 ± 0.70 | 4.13 ± 0.48 |
| *Interdisciplinary Humanities/Social Sciences* | 4.04 ± 0.74 | 4.11 ± 0.63 | 3.88 ± 0.77 | 4.13 ± 0.80 | 4.10 ± 0.74 |
| **Student Type:** *Undergraduate Students* | 4.01 ± 0.69 | 4.14 ± 0.65 | 3.77 ± 0.82 | 4.23 ± 0.74 | 4.14 ± 0.64 |
| *Graduate Students* | 3.95 ± 0.80 | 4.20 ± 0.57 | 3.62 ± 0.79 | 4.07 ± 0.84 | 4.05 ± 0.81 |

SD = Standard Deviation; DT = Design Thinking

Principal Component Analysis with Varimax rotation using Kaiser Normalization converged in 9 iterations and accounted for 63.01% of variance; no significant differences found between groups.

*Underrepresented race/ethnicity includes Black/African American, Hawaiian/Pacific Islander, American Indian/Alaskan, and Hispanic

differences in DT-TL approaches (e.g., business vs humanities). Additional research is needed to understand how and why DT is experienced differently by various stakeholders, and which organizational aspects of these contexts might mediate or support relevant outcomes.

## DT-TL practices

Although participants indicated frequently utilizing all DT practices, some were used more than others. Namely, participants indicated engaging in *Team Formation and Functioning* most frequently, which aligns with previous studies [21, 24] and likely reflects DT's commitment to collaborative project-based problem-solving. In contrast, participants engaged in *Prototyping and Experimentation* least frequently, which also aligns with previous studies in higher education [21]. We speculate that a lack of prototyping and experimentation could be a byproduct of dominant approaches to classroom learning that deemphasize the need for

**Table 5.** Side-by-side comparison of DT outcome factors in higher education and DT outcome factors in for-profit, non-profit, and Gov't settings.

| Current Study: Higher Education | Liedka and Bahr: For-Profit, Non-Profit, and Gov't [24] |
|---|---|
| **Factor 1: Implementation Support** | **Factor 1: Improved Implementation and Adaptation** |
| • *Improved the likelihood of the implementation of new solutions*<br>• *Encouraged shifts in organizational culture that made it more customer-focused*<br>• *Encouraged changes in organizational culture that made risk-taking more acceptable*<br>• *Broadened organization's definition of what innovation is*<br>• Expanded access to new resources for individuals and teams<br>• Created a sense of safety to try new things<br>• Helped people interested in trying new things to connect and support each other<br>• Encouraged people's open-mindedness to try new things | • *Improved the likelihood of the implementation of new solutions*<br>• *Encouraged shifts in organizational cultural that made it more customer-focused*<br>• *Encouraged changes in organizational culture that made risk-taking more acceptable*<br>• *Broadened organization's definition of what innovation is*<br>• Enhanced your ability to pivot when initial solution didn't work<br>• Made it easier to discard solutions that didn't work as planned<br>• Kept people motivated to work on a project to achieve impact<br>• Increase a sense of ownership and acceptance of a solution<br>• Increased appreciation for use of data to help drive decisions |
| **Factor 2: Psychological Benefits and Motivation** | **Factor 2: Individual Psychological Benefits** |

(*Continued*)

**Table 5.** (Continued)

| Current Study: Higher Education | Liedka and Bahr: For-Profit, Non-Profit, and Gov't [24] |
|---|---|
| • *Gave people more confidence in their own creative abilities*<br>• Kept people motivated to work on a project to achieve impact<br>• Increase a sense of ownership and acceptance of a solution<br>• Increased appreciation for use of data to help drive decisions<br>• Increased engagement of teammates involved in the design thinking process | • *Gave employees more confidence in their own creative abilities*<br>• Created a sense of safety to try new things<br>• Helped people interested in trying new things to connect and support each other<br>• Encouraged people's open-mindedness to try new things |
| **Factor 3: Relationships and Trust** | **Factor 3: Expanded Network Relationships and Resources** |
| • *Built new relationships locally that continued after the initial project was completed*<br>• *Enhanced other stakeholders willingness to collaborate on new solutions*<br>• Built trust among team members<br>• Built trust between problem-solving teams and other stakeholders<br>• Helped people involved to examine their own biases and preconceptions | • *Built new relationships locally that continued after the initial project was completed*<br>• *Enhanced other stakeholders' willingness to collaborate on new solutions*<br>• Expanded access to new resources for individuals and teams<br>• Helped pool resources for greater impact |
| **Factor 4: Quality of Solutions Generated** | **Factor 4: Quality of Solutions Generated** |
| • *Allowed new and better solutions, not visible at the beginning of the process, to emerge during it*<br>• *Fostered the inclusion of user input* | • *Allowed new and better solutions, not visible at the beginning of the process, to emerge during it*<br>• *Fostered the inclusion of user input*<br>• Helped teams see the problems in new ways, resulting in solving more promising problem<br>• Increased engagement of employees involved in the Design thinking process<br>• Helped people involved to examine their own biases and preconceptions |
| **Factor 5: Individual Adaptation and Flexibility** | **Factor 5: Trust Building** |
| • Helped me see the problems in new ways, resulting in solving more promising problems<br>• Enhanced my ability to pivot when initial solutions didn't work<br>• Made it easier to discard solutions that didn't work as planned | • Built trust among team members<br>• Built trust between problem-solving teams and other stakeholders |

DT = Design Thinking; Gov't = Government; *Italicized items indicate the same items were within the same factor for both studies.*

experiential and experimental practices and the constraints of a semester-long course (i.e., limited time available to iterate through the full DT process). While there are many student benefits to experiential learning [29], applied learning [30], process-based learning [31], and service-learning [32], they are time-consuming and resource intensive to execute and come with their own limitations and challenges [32]. We recommend faculty consider generating lower-stakes, quicker-paced student learning opportunities to prototype and test in addition to offering consecutive semester-long courses to ensure students are provided opportunities to develop these skills.

Compared to research in workplace settings [24], our faculty and students reported a greater frequency of the following front-end practices: *followed a structured process, emphasized active listening among team to find shared meaning,* and *focused problem definition on the*

**Table 6. Bivariate correlations (r_p) and reliabilities (α, in parentheses) for DT constructs.**

|  | 1 | 2 | 3 | 4 | 5 | 6 | 7 | 8 |
|---|---|---|---|---|---|---|---|---|
| 1. Discovery & Ideation | (0.76) |  |  |  |  |  |  |  |
| 2. Team Formation & Functioning | 0.51 | (0.53) |  |  |  |  |  |  |
| 3. Prototyping & Experimentation | 0.39 | 0.36 | (0.77) |  |  |  |  |  |
| 4. Implementation Support | 0.46 | 0.60 | 0.45 | (0.89) |  |  |  |  |
| 5. Psychological Benefits & Motivation | 0.41 | 0.52 | 0.33 | 0.76 | (0.84) |  |  |  |
| 6. Relationships & Trust | 0.33 | 0.51 | 0.44 | 0.73 | 0.65 | (0.82) |  |  |
| 7. Quality of Solutions Generated | 0.52 | 0.52 | 0.42 | 0.58 | 0.53 | 0.53 | (0.68) |  |
| 8. Individual Adaptation & Flexibility | 0.47 | 0.46 | 0.34 | 0.66 | 0.57 | 0.54 | 0.47 | (0.74) |

p < .05 for all correlations

*user's perspective rather than the organization's.* On the other hand, some practices associated with the later phase of the DT process were reported at lower frequency by faculty and students in our study compared to participants in the Liedtka and Bahr [24] study, including: *moved multiple ideas into prototyping, got feedback form users and other stakeholders on prototype*, and *executed real world experiments.*

## Outcomes of DT-TL

Recent research suggests that DT can empower students to design desirable, feasible, transdisciplinary solutions that promote practical and sustainable outcomes [22, 33]. Our results align with Lake and colleagues [22], who also found that the *Quality of Solutions Generated* was the most frequently experienced outcome, highlighting the process of engaging students in the DT process as a means for solving complex problems. Not surprisingly, *Quality* as an outcome had a moderately strong, positive relationship with the practice of *Team Formation and Functioning*, highlighting the potential benefits of well-designed, high-structured teamwork within courses.

In addition, literature highlights potential cognitive and behavioral benefits of DT, showing positive cognitive and behavioral changes for learning and decision-making [28, 34]. Our results align with this literature, suggesting that participants frequently experienced *Psychological Benefits and Motivation* (e.g., *kept people motivated to work on a project to achieve impact*). These findings are crucial for educators seeking evidence that DT teaching practices provide students with skills and mindsets for more inclusively and resiliently addressing complex, real world challenges. Both benefits highlight the importance of establishing relevance between student learning and real-world situations through DT pedagogies.

We posit that our outcomes of DT-TL were similar and different from those of Liedtka and Bahr [24] due to our different study populations and contexts. We adapted the survey instrument for faculty and students in higher education institutions involved in semester-long classes whereas Liedtka and Bahr [24] designed it for employees of for-profit, non-profit, and government entities. In our sample, the items on trust (*built trust among team members, built trust between problem-solving teams and other stakeholders*) loaded alongside other relationship items whereas, in Liedtka and Bahr [24], those items loaded onto its own factor. In academic settings, we often talk about relationship-building and trust together, especially as we discuss community engagement, so this makes sense. In community-based participatory research in particular, relationship-building and trust-building go hand-in-hand [35, 36].

We also found that survey items *kept people motivated to work on a project to achieve impact, increased a sense of ownership and acceptance of a solution*, and *increased appreciation*

*for use of data to help drive decisions* loaded with the factor on psychological benefits rather than the improved implementation and adaptation factor from Liedtka and Bahr [24]. These three items mark shifts in individual mindsets so this grouping makes sense. Relatedly, the new factor we proposed is termed "Individual Adaptation and Flexibility" and centers around growth mindset and a willingness to learn and change.

## DT-TL group differences

Our findings generally indicate that DT-TL practices and valued outcomes are prevalent across disciplines, providing early insights into the potential merits of DT-TL as an interdisciplinary process that is valued across institutions. Several group comparisons warrant discussion and further research, such as undergraduate students experiencing *Prototyping and Experimentation* more frequently than graduate students. Although graduate degree programs typically immerse graduate students in research, we posit that the structure and processes within graduate programs–which tend to include close oversight from an expert faculty member in a focused area of study–may limit the creativity and brainstorming opportunities afforded to undergraduates. Alternatively, if graduate students experience Prototyping and Experimentation frequently in their degree program, they may perceive this practice as less frequent in DT coursework compared to undergraduate students who are not engaged in similar research programs. As it relates to race and ethnicity, lack of differences should be interpreted with caution since data were collected from predominantly white institutions and the sample sizes for subgroups were relatively small. However, this finding should be explored further given legitimate concerns that DT practices can further privilege those with privileged identities [37, 38].

## Future directions

This study marks an important step in understanding DT across institutions and disciplines within higher education *and* across educational and professional divides. Our findings, coupled with those from K-12 education–where DT is lauded as an integral cognitive process involving creation, collaborative sense-making, reasoning with evidence, experimentation, and evaluation [39–41]–pose several intriguing opportunities for next steps.

Taken together, K-12, higher education, postgraduate, and professional studies contribute to the growing body of research that demonstrates clear DT benefits for students throughout their educational careers and beyond. As education faces ongoing scrutiny about its relevance and value, educators must adopt strategies that enable students to address increasingly complex real-world challenges. In *Doctors as Makers*, for example, Baruch implores medical curricula to foster creative and critical thinkers that can work through not-knowing, seek compassionate solutions, and explore questions differently within an environment supportive of iterative development [42]. Understanding the ways in which DT compliments traditional problem-solving frameworks, such as the scientific method and clinical decision making, could be an important step toward optimizing how DT-TL is integrated into our curricula [16].

With its emphasis on situated and relational problem-solving frameworks, DT-TL also aligns with "problem-posing" pedagogical theory and praxis that is core to various academic disciplines, such as innovation and entrepreneurship. In *Pedagogy of the Oppressed*, Freire described problem-posing education as a shift from a content-centric, hierarchical educational model to a contextually responsive, co-creative one [43]. Specifically, DT-TL maps seamlessly onto many of the problem-posing processes taught in innovation and entrepreneurship education, particularly to the discovery process (e.g., *problem identification through understanding stakeholder needs*) and the development process (e.g., *iterative testing and evaluation of*

*prototypes*). Similarly, other pedagogical models, such as community-based, project-based, and problem-based-learning include components that align closely with DT. Since institutions who make DT a central strategic focus are more likely to bolster and sustain a competitive advantage [44], more research is needed in higher education to understand the potential prevalence and relationship of DT-TL to pedagogical praxis and other pedagogical models (e.g., problem-based learning) across various disciplines.

Moving forward, we encourage DT-TL educators to extend our work by collecting similar data, developing improved quantitative measures, and implementing longitudinal studies with faculty, students, and project partners. Larger, more representative samples will promote generalizability and enable analyses and sub-analyses examining additional institution- (e.g., university size, Carnegie Classification, institutional control) and faculty-level (e.g., number of years teaching DT, discipline of home department or school, rank or track) factors. Moreover, it would be helpful to know how often students experience DT prior to taking these courses, shortly after taking these courses, and several months or years after taking the courses. This could allow us to better establish causality and attribute changes to the specific courses. Additional research should also be conducted to understand DT-TL outcomes for co-curricular opportunities, such as student organizations and internships to determine what other opportunities in higher education settings can yield similar or better outcomes.

## Limitations

With regards to study limitations, we encountered standard challenges with response rates, missing data, and non-response. We recruited faculty and students from four predominantly White higher education institutions in the same state and had a small sample size of faculty at each institution. Data also reflect student and faculty perception at the end of a course–not the long-term outcomes/value of these practices for them or for their project collaborators. That being said, our results are not meant to be generalizable to all DT courses in across higher education settings. Rather, they provide an important first glimpse into DT-TL practices across multiple universities and disciplines.

## Conclusion

College students must be equipped with the critical, creative, and collaborative skills needed to address society's increasingly complex and difficult challenges. As this study further verified, DT helps to generate trust across collaborators, fosters the motivation needed to sustain problem-solving efforts, and increases the quality of solutions generated. This study demonstrated the validity of DT across disciplines and universities, the ways in which DT practices and outcomes are experienced in higher education, and a number of important differences between DT practices within higher education and other sectors. Extending this work to include additional universities and research methodologies is critical for providing insight into enhancing the value of DT pedagogies.

## Supporting information

**S1 Table. Outcomes of design thinking coursework.**
(DOCX)

## Acknowledgments

The authors would like to acknowledge the faculty and students who participated in the study.

## Author Contributions

**Conceptualization:** Jacqueline E. McLaughlin, Elizabeth Chen, Danielle Lake, Wen Guo, Aria Chernik, Tsailu Liu.

**Data curation:** Jacqueline E. McLaughlin.

**Formal analysis:** Jacqueline E. McLaughlin, Wen Guo, Emily Rose Skywark.

**Investigation:** Jacqueline E. McLaughlin, Elizabeth Chen, Danielle Lake.

**Methodology:** Jacqueline E. McLaughlin, Elizabeth Chen, Danielle Lake, Aria Chernik, Tsailu Liu.

**Project administration:** Jacqueline E. McLaughlin, Elizabeth Chen, Danielle Lake, Aria Chernik.

**Supervision:** Danielle Lake.

**Validation:** Jacqueline E. McLaughlin.

**Writing – original draft:** Jacqueline E. McLaughlin, Elizabeth Chen, Danielle Lake, Emily Rose Skywark.

**Writing – review & editing:** Jacqueline E. McLaughlin, Elizabeth Chen, Danielle Lake, Wen Guo, Emily Rose Skywark, Aria Chernik, Tsailu Liu.

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
