## [Decision Letter · Decision Letter 0]

19 Jan 2022

PONE-D-21-32075Design Thinking Teaching and Learning in Higher Education: Student and Faculty Experiences Across Four UniversitiesPLOS ONE

Dear Dr. McLaughlin,

Thank you for submitting your manuscript to PLOS ONE. After careful consideration, we feel that it has merit but does not fully meet PLOS ONE’s publication criteria as it currently stands. Therefore, we invite you to submit a revised version of the manuscript that addresses the points raised during the review process.

We look forward to receiving your revised manuscript.

Kind regards,

Oathokwa Nkomazana, MD MSC PhD

Academic Editor

PLOS ONE

Journal Requirements:

2. 1. Please ensure that you include a title page within your main document. We do appreciate that you have a title page document uploaded as a separate file, however, as per our author guidelines (http://journals.plos.org/plosone/s/submission-guidelines#loc-title-page) we do require this to be part of the manuscript file itself and not uploaded separately.

Reviewers' comments:

Reviewer's Responses to Questions

**Comments to the Author**

1. Is the manuscript technically sound, and do the data support the conclusions?

Reviewer #1: Yes

Reviewer #2: Yes

2. Has the statistical analysis been performed appropriately and rigorously? 

Reviewer #1: I Don't Know

Reviewer #2: Yes

3. Have the authors made all data underlying the findings in their manuscript fully available?

Reviewer #1: Yes

Reviewer #2: Yes

4. Is the manuscript presented in an intelligible fashion and written in standard English?

Reviewer #1: Yes

Reviewer #2: Yes

5. Review Comments to the Author

Reviewer #1: This is a very important manuscript regarding DT-TL. As authors have stated, this is a subject that taught mainly in certain disciplines and not in other. The authors therefore have shared pioneering work in regard to DT in Humanities/ Social Sciences. The introduction is clearly written. There are a number of queries that I have noted that may improve the quality of the manuscript:

1. The authors could also include and infer about the impact of DT in various curricula in their introduction or discussion since they mention that DT is already taught in certain disciplines. The authors could also highlight the type of curricula for the four universities and/or courses of the participants. Certain curricula have some components of the DT such as Problem Based Learning curriculum.

2. They adapted a survey from Liedtka and Bahr who studied DT practices among the workforces. The authors were not clear how after adaptation they validated the tool before they started their research.

3. In their research questions, they could have grammatic review and edit the questions to make them more clearer and highlight the sub-questions.

4. There are sentences where a semi-colon is more appropriate than a comma or a hyphen (line 96)

5. With regard to ethical consideration, the authors obtained 3 IRB approvals, but 4 universities participated. They remain silent on the fourth university IRB approval.

6. Within the results section, the authors mention that the majority of participants majored in Humanities/ Social Sciences and that they were mainly white and undergraduates. Since the authors discusses these variables in the results/discussion, it remains succinctly prudent for them to mention all the disciplines, non-white population participating in the study and, the number of graduate students involved. Is discussing these variables in place in terms of population size of each?

7. Should the standard deviations be written without a zero (.68) or, in full starting with a zero (0.68)? If so, it should be consistently added to all the standard deviations.

8. In the discussion, they could briefly share their thoughts of some faculty who teach DT without the DT expertise.

9. In the discussion, the authors mention how their research complements conceptual frameworks of DT in education: they could mention two or three of them.

10. Write org in full (line 243)

11. Line 287: the authors could share their thoughts on the reason why undergraduate students reported Prototyping and Experimentation more frequently than graduate students as the reverse is expected as the norm with graduate.

12. The authors decided to publish the qualitative component of the research separately, some of the critical perspectives about DT-TL may be missing in this manuscript especially that it seems the authors have not fully answered some of the research questions such as the No.3 research question (Is DT a valid construct within teaching and learning?)

13. The section under Future Directions seems to be too long and at times reads more like the general discussion.

14. They should be consistent with their referencing

Reviewer #2: Design Thinking Teaching and Learning in Higher Education: Student and Faculty Experiences Across Four Universities

Summary observations

Abstract is clear although the results component is in places inconsistent with what is indicated in the main results section of the manuscript. In the introduction, the authors indicate that ‘DT practices and outcomes across higher education is still new’. Therefore, they did not discuss or reference any available published literature in the ‘introduction’, to give a sense of the overall context. This creates the impression that nothing has been published in this area. The specific research questions are well-articulated. The researchers should have stated the main research question before listing the specific research questions. Appropriate references are quoted, institutional review boards’ approval were obtained and informed consent obtained. Tables are clear and readable but Table 1 needs to be revised to reflect ‘missing data’. The data were collected, analysed and interpreted by and large correctly. However, the results that reflect the ‘main findings’ of the study as stated in the abstract should have been presented in a table to enable the reviewers to interrogate the data analysis and the conclusion thereof. This was not possible. The results are adequately discussed but there are instances where the key findings are stated with little or no discussion. The authors discuss the limitation of the study. However, some of the limitations mentioned are not limitation per se. Language editing is strongly recommended.

Specific Comments

Abstract

L15/16 contradicts/inconsistent with L287. ‘Differences found based on discipline and students’ level’. Pay attention to statement on ‘student level’

Introduction

L26-29 Long sentence

L48 – ‘wicked’ not sure it is appropriate for formal writing

L66 – “outcomes across higher education is still new’. Though ‘new’ what does the limited/available

published data show? and give references.

L73-81. Are specific research questions but what is the main/overall research question? State it

first before giving specific research questions?

Data Collection

L101-104 has to be reconciled with L152/153 (Results section). Purposive sampling was used and faculty who teach design thinking courses were recruited (L101-L104). If this is true, how come faculty with ‘none’ DT expertise (L152) were recruited? How does this help answer the research questions?

Ethical considerations, consent

L132-134. Study involved four (4) universities (L100) but only three (3) review boards are mentioned.

Results

L159/160. It is ‘shared meeting’ or ‘shared meaning’?

L170-175. Reading the abstract it appears L170-175 represent the ‘main findings’ of the study. How come then the analysis is not presented in a table to enable the reviewers to interrogate the data and the analysis. Suggest that the results be tabulated.

Discussion

L209. ‘First and foremost’ unnecessary words

L213. ‘8 outcomes constructs’ shouldn’t it be ‘5 outcome constructs’

L236-242. It is simply reporting the results and there is no discussion; should be moved to either the results section or inserted after line 226

L243. ‘Org.’ should be written in full.

L266-267. Please revise to connect the two sentences better

L283. ‘centers around growth mindset and a willingness to learn and change’. Revise this sentence, not easy to understand. Should it be ‘centers around (on) growth mindset (growth of mindset) and a willingness to learn and change ‘

L290/291. Needs elaborating and clarifying. Can this conclusion be drawn?, given that the study was conducted in predominantly white institutions and the numbers for other races (individually) are low; much underrepresented.

Future Directions

L310. Is it ‘institution who…’?

L316-328. Not sure about the relevance of these lines to the manuscript. Looks like it is referring to the ‘qualitative component of the overall study’ which is not the subject of this manuscript.

Limitations

L358. Reconcile this with L85-87. How can ‘lack of qualitative data’ be a limitation when this was a mixed methods study and the researchers ‘intentionally’ decided not to consider the qualitative and quantitative data together?

Conclusion

L368-370. Please revise. It is unclear and incomplete. For instance, what is “promote mindsets…’?

Table 1

• The students’ numbers do not add up to 196 and percentages do not up to 100% for the participants’ characteristics. Please include in the table ‘missing/unknowns/no responses’ in your table.

• Why is it that only one ethnicity is recorded? It is also not clear what its relevance is.

6. PLOS authors have the option to publish the peer review history of their article (what does this mean?). If published, this will include your full peer review and any attached files.

Reviewer #1: No

Reviewer #2: **Yes: **Enoch Sepako, PhD, MHPE

---

## [Author Response · Author response to Decision Letter 0]

18 Feb 2022

The following responses are also included as a separate file in this resubmission:

Academic Editor Comments: 

Manuscript and file names revised to meet style requirements

2. 1. Please ensure that you include a title page within your main document. We do appreciate that you have a title page document uploaded as a separate file, however, as per our author guidelines (http://journals.plos.org/plosone/s/submission-guidelines#loc-title-page) we do require this to be part of the manuscript file itself and not uploaded separately.

Title page added to the manuscript.

2. Please include your tables as part of your main manuscript and remove the individual files. Please note that supplementary tables (should remain/ be uploaded) as separate "supporting information" files

Tables added to the main manuscript directly after the paragraph in which each is first cited.

As noted in the cover letter, we have made our data available in ICPSR at https://doi.org/10.3886/E151681V1 (ie the DOI for our data is 10.3886/E151681V1)

References updated per style guidelines. No references were removed or added.

Reviewer Comments to the Author

Reviewer #1 

This is a very important manuscript regarding DT-TL. As authors have stated, this is a subject that taught mainly in certain disciplines and not in other. The authors therefore have shared pioneering work in regard to DT in Humanities/ Social Sciences. The introduction is clearly written. 

Thank you for this feedback. We have addressed your recommendations below and believe this has strengthened our work.

There are a number of queries that I have noted that may improve the quality of the manuscript:

1. The authors could also include and infer about the impact of DT in various curricula in their introduction or discussion since they mention that DT is already taught in certain disciplines. The authors could also highlight the type of curricula for the four universities and/or courses of the participants. Certain curricula have some components of the DT such as Problem Based Learning curriculum.

Thanks for this comment. As noted in Table 1, we collected some data about course characteristics (e.g., Real World Project Used for DT, Percent of Course Time Spent in Teams). However, we did not collect any other details or data about the curricula/pedagogy for this study (such as PBL). In the discussion, we have revised our paragraph about innovation and entrepreneurship training in higher education to address the need to explore the impact DT more closely as it relates to the various disciplines and pedagogies (e.g., PBL). 

2. They adapted a survey from Liedtka and Bahr who studied DT practices among the workforces. The authors were not clear how after adaptation they validated the tool before they started their research.

This paper provides validity evidence for the revised tool. Prior to administering the survey, we used face validity to adapt the tool, as now noted in the methods. After administering the survey, we used factor analysis and correlation analysis to generate validity evidence.

3. In their research questions, they could have grammatic review and edit the questions to make them more clearer and highlight the sub-questions.

Research questions and sub-questions revised for clarity.

4. There are sentences where a semi-colon is more appropriate than a comma or a hyphen (line 96)

Punctuation revisions made where considered appropriate by the authorship team.

5. With regard to ethical consideration, the authors obtained 3 IRB approvals, but 4 universities participated. They remain silent on the fourth university IRB approval.

Revised to include the fourth review board.

6. Within the results section, the authors mention that the majority of participants majored in Humanities/ Social Sciences and that they were mainly white and undergraduates. Since the authors discusses these variables in the results/discussion, it remains succinctly prudent for them to mention all the disciplines, non-white population participating in the study and, the number of graduate students involved. Is discussing these variables in place in terms of population size of each?

Statistics for all demographic variables collected in this study are included in Table 1. We have chosen not to write all of these results in the text, since it would be duplicative of the Table. 

7. Should the standard deviations be written without a zero (.68) or, in full starting with a zero (0.68)? If so, it should be consistently added to all the standard deviations.

Leading zeros have been added to all data, including standard deviations.

8. In the discussion, they could briefly share their thoughts of some faculty who teach DT without the DT expertise.

All faculty who teach DT indicated having some DT expertise, as noted in Table 1. We have clarified this point in the results.

9. In the discussion, the authors mention how their research complements conceptual frameworks of DT in education: they could mention two or three of them.

Two examples added.

10. Write org in full (line 243)

Revised. 

11. Line 287: the authors could share their thoughts on the reason why undergraduate students reported Prototyping and Experimentation more frequently than graduate students as the reverse is expected as the norm with graduate.

This has been addressed in the discussion section, which posits that graduate degree programs typically immerse graduate students in research– typically with close oversight from an expert faculty member in a focused area of study – that may limit the creativity and brainstorming opportunities afforded to undergraduates. Alternatively, if graduate students experience Prototyping and Experimentation frequently in their degree program, they may perceive this practice as less frequent in DT coursework compared to undergraduate students who are not engaged in similar research programs.

12. The authors decided to publish the qualitative component of the research separately, some of the critical perspectives about DT-TL may be missing in this manuscript especially that it seems the authors have not fully answered some of the research questions such as the No.3 research question (Is DT a valid construct within teaching and learning?)

The factor analysis conducted in this study provides strong construct validity evidence for DT-TL. As such, we argue that DT-TL is a valid construct based on this evidence. This is described in detail in the Discussion: DT-TL Constructs section. 

13. The section under Future Directions seems to be too long and at times reads more like the general discussion.

This section has been shortened and revised so that each paragraph concludes with a next step/future direction.

14. They should be consistent with their referencing.

References were revised for according to journal style requirements for consistency.

Reviewer #2

Summary observations

Abstract is clear although the results component is in places inconsistent with what is indicated in the main results section of the manuscript. In the introduction, the authors indicate that ‘DT practices and outcomes across higher education is still new’. Therefore, they did not discuss or reference any available published literature in the ‘introduction’, to give a sense of the overall context. This creates the impression that nothing has been published in this area. The specific research questions are well-articulated. The researchers should have stated the main research question before listing the specific research questions. Appropriate references are quoted, institutional review boards’ approval were obtained and informed consent obtained. Tables are clear and readable but Table 1 needs to be revised to reflect ‘missing data’. The data were collected, analysed and interpreted by and large correctly. However, the results that reflect the ‘main findings’ of the study as stated in the abstract should have been presented in a table to enable the reviewers to interrogate the data analysis and the conclusion thereof. This was not possible. The results are adequately discussed but there are instances where the key findings are stated with little or no discussion. The authors discuss the limitation of the study. However, some of the limitations mentioned are not limitation per se. Language editing is strongly recommended.

Thank you for this feedback. We have addressed your observations below in point-by-point responses and believe these revisions have strengthened our work.

Specific Comments

Abstract

L15/16 contradicts/inconsistent with L287. ‘Differences found based on discipline and students’ level’. Pay attention to statement on ‘student level’

Corrected in abstract. “Student level” revised to “student type” throughout.

Introduction

L26-29 Long sentence

Sentence shortened.

L48 – ‘wicked’ not sure it is appropriate for formal writing

Word removed. 

L66 – “outcomes across higher education is still new’. Though ‘new’ what does the limited/available

published data show? and give references.

To our knowledge, ours is the first study to collect and validate data across multiple higher education institutions. We have expanded this paragraph to note what others have found and that their studies have been implemented at a single institution or single course.

L73-81. Are specific research questions but what is the main/overall research question? State it

first before giving specific research questions?

The overarching research question has been added (how do faculty and students experience design thinking within higher education courses?), prior to giving the specific research questions.

Data Collection

L101-104 has to be reconciled with L152/153 (Results section). Purposive sampling was used and faculty who teach design thinking courses were recruited (L101-L104). If this is true, how come faculty with ‘none’ DT expertise (L152) were recruited? How does this help answer the research questions?

As seen in Table 1, all faculty reported have at least limited expertise (none of them indicated having “none”). This has been clarified in the Results section.

Ethical considerations, consent

L132-134. Study involved four (4) universities (L100) but only three (3) review boards are mentioned.

Revised to include the fourth review board.

Results

L159/160. It is ‘shared meeting’ or ‘shared meaning’?

Corrected to “shared meaning”

L170-175. Reading the abstract it appears L170-175 represent the ‘main findings’ of the study. How come then the analysis is not presented in a table to enable the reviewers to interrogate the data and the analysis. Suggest that the results be tabulated.

Demographic results added to Tables 3 & 4

Discussion

L209. ‘First and foremost’ unnecessary words

Removed.

L213. ‘8 outcomes constructs’ shouldn’t it be ‘5 outcome constructs’

Revised.

L236-242. It is simply reporting the results and there is no discussion; should be moved to either the results section or inserted after line 226

Section removed.

L243. ‘Org.’ should be written in full.

Revised.

L266-267. Please revise to connect the two sentences better

Revised.

L283. ‘centers around growth mindset and a willingness to learn and change’. Revise this sentence, not easy to understand. Should it be ‘centers around (on) growth mindset (growth of mindset) and a willingness to learn and change ‘

Removed “growth mindset” and revised to “since it embodies…”

L290/291. Needs elaborating and clarifying. Can this conclusion be drawn?, given that the study was conducted in predominantly white institutions and the numbers for other races (individually) are low; much underrepresented.

Revised to acknowledge low sample size and predominantly white institutions. 

Future Directions

L310. Is it ‘institution who…’?

Revised.

L316-328. Not sure about the relevance of these lines to the manuscript. Looks like it is referring to the ‘qualitative component of the overall study’ which is not the subject of this manuscript.

This section has been revised to align more clearly with the current study.

Limitations

L358. Reconcile this with L85-87. How can ‘lack of qualitative data’ be a limitation when this was a mixed methods study and the researchers ‘intentionally’ decided not to consider the qualitative and quantitative data together?

Limitation removed.

Conclusion

L368-370. Please revise. It is unclear and incomplete. For instance, what is “promote mindsets…’?

Removed. 

Table 1

• The students’ numbers do not add up to 196 and percentages do not up to 100% for the participants’ characteristics. Please include in the table ‘missing/unknowns/no responses’ in your table.

A note has been added to Table 1 to clarity that some variables have missing data.

• Why is it that only one ethnicity is recorded? It is also not clear what its relevance is.

The use of this ethnicity is standard practice for demographic data collection in the United States. Per the US Census Bureau “Though many respondents expect to see a Hispanic, Latino, or Spanish category on the race question, this question is asked separately because people of Hispanic origin may be of any race(s).”

---

## [Decision Letter · Decision Letter 1]

10 Mar 2022

Design Thinking Teaching and Learning in Higher Education: Experiences Across Four Universities

PONE-D-21-32075R1

Dear Dr. McLaughlin,

We’re pleased to inform you that your manuscript has been judged scientifically suitable for publication and will be formally accepted for publication once it meets all outstanding technical requirements.

Kind regards,

Alessandro Margherita

Academic Editor

PLOS ONE

Additional Editor Comments (optional):

Reviewers' comments:

Reviewer's Responses to Questions

**Comments to the Author**

1. If the authors have adequately addressed your comments raised in a previous round of review and you feel that this manuscript is now acceptable for publication, you may indicate that here to bypass the “Comments to the Author” section, enter your conflict of interest statement in the “Confidential to Editor” section, and submit your "Accept" recommendation.

Reviewer #1: All comments have been addressed

Reviewer #2: All comments have been addressed

2. Is the manuscript technically sound, and do the data support the conclusions?

Reviewer #1: Yes

Reviewer #2: Yes

3. Has the statistical analysis been performed appropriately and rigorously? 

Reviewer #1: I Don't Know

Reviewer #2: Yes

4. Have the authors made all data underlying the findings in their manuscript fully available?

Reviewer #1: Yes

Reviewer #2: Yes

5. Is the manuscript presented in an intelligible fashion and written in standard English?

Reviewer #1: Yes

Reviewer #2: Yes

6. Review Comments to the Author

Reviewer #1: The authors have sufficiently responded to reviewers comments. The authors mentioned publishing the qualitative component of the research separately. This part of work should be discussed only if it adds-value to this manuscript discussions.

Reviewer #2: L103 -L106: 'In one single institution study, for example, researchers found that DT requires time and trust which can be constrained by the imposed deadlines of semester-based projects [21]. In a single course study, students indicated that their “whirlwind” course promoted almost “exponential” growth [22]' should be inserted after L108.

7. PLOS authors have the option to publish the peer review history of their article (what does this mean?). If published, this will include your full peer review and any attached files.

Reviewer #1: No

Reviewer #2: **Yes: **Enoch Sepako, PhD

---

## [Editor Report · Acceptance letter]

15 Mar 2022

PONE-D-21-32075R1 

Design thinking teaching and learning in higher education:
Experiences across four universities 

Dear Dr. McLaughlin:

I'm pleased to inform you that your manuscript has been deemed suitable for publication in PLOS ONE. Congratulations! Your manuscript is now with our production department. 

Kind regards, 

on behalf of

Dr. Alessandro Margherita 

Academic Editor

PLOS ONE